# Retinoic Acid-Containing Liposomes for the Induction of Antigen-Specific Regulatory T Cells as a Treatment for Autoimmune Diseases

**DOI:** 10.3390/pharmaceutics13111949

**Published:** 2021-11-17

**Authors:** Daniëlle ter Braake, Naomi Benne, Chun Yin Jerry Lau, Enrico Mastrobattista, Femke Broere

**Affiliations:** 1Department of Infectious Diseases and Immunology, Faculty of Veterinary Medicine, Utrecht University, 3584 CL Utrecht, The Netherlands; d.terbraake@uu.nl (D.t.B.); n.benne@uu.nl (N.B.); 2Department of Pharmaceutics, Utrecht Institute for Pharmaceutical Sciences, Faculty of Science, Utrecht University, 3584 CG Utrecht, The Netherlands; c.y.lau@uu.nl (C.Y.J.L.); e.mastrobattista@uu.nl (E.M.)

**Keywords:** tolerance, liposomes, nanoparticles, tolerogenic dendritic cells, retinoic acid, autoimmunity

## Abstract

The current treatment of autoimmune and chronic inflammatory diseases entails systemic immune suppression, which is associated with increased susceptibility to infections. To restore immune tolerance and reduce systemic side effects, a targeted approach using tolerogenic dendritic cells (tolDCs) is being explored. tolDCs are characterized by the expression of CD11c, the major histocompatibility complex (MHC)II and low levels of co-stimulatory molecules CD40 and CD86. In this study, tolDCs were generated using a human-proteoglycan-derived peptide (hPG) and all-trans retinoic acid (RA). RA-tolDCs not only display a tolerogenic phenotype but also can induce an antigen-specific regulatory T cell (Treg) response in vitro. However, further analysis showed that RA-tolDCs make up a heterogeneous population of DCs, with only a small proportion being antigen-associated tolDCs. To increase the homogeneity of this population, 1,2-distearoyl-*sn*-glycero-3-phosphoglycerol (DSPG)-containing liposomes were used to encapsulate the relevant antigen together with RA. These liposomes greatly enhanced the proportion of antigen-associated tolDCs in culture. In addition, in mice, we showed that the liposomal co-delivery of antigen and RA can be a more targeted approach to induce antigen-specific tolerance compared to the injection of RA-tolDCs, and that these liposomes can stimulate the generation of antigen-specific Tregs. This work highlights the importance of the co-delivery of an antigen and immunomodulator to minimize off-target effects and systemic side effects and provides new insights in the use of RA for antigen-specific immunotherapy for autoimmune and chronic inflammatory diseases.

## 1. Introduction

Autoimmune diseases and chronic inflammatory diseases are major public health concerns in Europe [1]. As there is no cure for these diseases, patients often require life-long treatment with immune-suppressing medication, which may be accompanied by severe side effects. In addition, the use of immunosuppressive drugs can increase the risk of infection [2]. Therefore, there is a great need to develop more effective treatments for autoimmune and chronic inflammatory diseases. In several autoimmune disorders, an imbalance in immune homeostasis is observed. This imbalance can be attributed to a loss of function or the reduced presence of antigen-specific suppressive immune cells, resulting in a breach of immune tolerance [3]. Immune tolerance is generally maintained by a variety of immune cells, including subsets of dendritic cells (DCs), T and B cells [4,5,6]. In autoimmune disorders, these cells recognize autoantigens as non-self and elicit a pro-inflammatory immune response. To date, several autoimmune disorders have been linked to specific autoantigens [7]. Antigen recognition is mediated by antigen-presenting cells (APCs), such as DCs. These cells continuously sense their environment through pattern recognition receptors on their cell surface [8]. Under inflammatory conditions, the detection of an antigen by these receptors causes DCs to become activated and migrate to draining lymph nodes. This dendritic cell maturation results in an upregulation of the antigen-presenting MHC molecules, chemokine receptors and an increase in pro-inflammatory cytokine secretion [9]. Naive T cells reside in the draining lymph nodes, where DCs can initiate effector T-cell responses through antigen presentation, co-stimulation and cytokine secretion [10]. However, some specialized sub-types of immune cells, such as tolerogenic dendritic cells (tolDCs), can help maintain immune tolerance. tolDCs are derived from immature dendritic cells upon encountering a tolerogenic stimulus and an activation stimulus [11,12]. These tolDCs can induce T-cell anergy, inhibit the proliferation of effector T cells (such as the pro-inflammatory CD4+ T helper subsets Th1, Th2 and Th17) and can promote regulatory T cell (Treg) differentiation [13]. tolDCs are characterized by a semi-mature phenotype, in which they show a reduced expression of co-stimulatory molecules (CD40, CD86) as compared to mature DCs (mDCs) on their cell surface and can secrete anti-inflammatory molecules, such as IL-10, to mediate immune suppression. Therefore, these cells are of great interest when developing treatments for autoimmune and chronic inflammatory diseases, in which immune tolerance needs to be restored. Previously, our lab and others have cultured tolDCs using a variety of immunomodulators, including dexamethasone and vitamin D3 [11]. In this study, we used all-*trans* retinoic acid (RA) for the culture of tolDCs, as has been described before [14,15,16].

RA is an active metabolite of vitamin A (all-*trans* retinol) that has been shown to play a significant role in the induction and maintenance of gut immune tolerance. The gut is host to a subpopulation of specialized DCs, which are able to metabolize food-derived vitamin A to RA [17]. RA can prime other gut-associated DCs to become RA-producing CD103^+^ DCs [18]. These DCs can subsequently convert naive T cells into Tregs [19,20,21]. This tolerance-inducing ability makes tolDCs interesting targets for the development of antigen-specific immunotherapy for autoimmune and chronic inflammatory diseases. Ex vivo culturing of patient DCs, converting them to tolDCs that present a disease-specific antigen and reinjecting them into the patient has already shown to be promising in clinical trials for several autoimmune diseases [22]. However, the process of isolating DC precursors from patients, stimulating the cells ex vivo and injecting them back into patients will remain not only labor intensive, but is also restricted to highly specialized cell culture facilities, thereby limiting the number of patients to be treated. Therefore, we propose that a delivery system such as nanoparticles can be a suitable alternative to tolDC culture. The use of nanoparticles, such as liposomes, shows great promise for in vivo immunomodulation, and liposomes have been widely used as a delivery vehicle for antigens and adjuvants [23,24]. In this study, we selected anionic 1,2-distearoyl-*sn*-glycero-3-phosphoglycerol (DSPG)-containing liposomes of around 200 nm in size as a delivery system, since these liposomes were shown to be inherently tolerogenic [25], and RA is lipophilic, so would be readily encapsulated into the lipid bilayer. To assess the antigen specificity of these nanocarriers, a mouse model in which the T cells of the animal only express T-cell receptors specific for human proteoglycan (hPG) was used. The hPG antigen has been widely used for the induction of proteoglycan-induced arthritis (PGIA) in mice [12], which is a model for autoimmunity. This study aimed to see whether liposomes can be suitable carriers for the hPG antigen and RA and if these liposomes are as effective in vivo as using antigen-specific RA tolDCs.

## 2. Materials and Methods

### 2.1. Synthesis of Peptides and Conjugates

Dimethylformamide, *N,N*′-diisopropyl carbodiimide, piperidine and acetonitrile were purchased from Biosolve BV, Valkenswaard, Netherlands. 9-fluorenylmethyloxycarbonyl (Fmoc)-protected amino acids, Fmoc-Lys (Boc)-Wang resin and trifluoroacetic acid were purchased from Novabiochem GmbH, Darmstadt, Germany. The peptide epitope sequences were synthesized by microwave-assisted solid-phase peptide synthesis using an H12 liberty blue peptide synthesizer (CEM Corporation, Stallings, NC, USA). Dimethylformamide was used as the coupling and washing solvent for the whole synthesis process. For each coupling step, Fmoc-protected amino acids were activated by five eq of Oxyma pure (Manchester Organics, Runcorn, UK) and *N,N*′-diisopropyl carbodiimide to react with the free *N*-terminal amino acids in preloaded resin for 1 min at 90 °C. After each coupling step, the Fmoc group was removed by treatment with 20% piperidine for 1 min at 90 °C. Fluorescein (FAM, ThermoFisher, Landsmeer, Netherlands) was coupled to the *N*-terminal of the peptide as with other Fmoc-protected amino acids. Trifluoroacetic acid/water/triisopropylsilane (Sigma-Aldrich, Zwijndrecht, Netherlands) (95/2.5/2.5) was used to simultaneously cleave the peptide off the resin and remove the side chain protecting groups. Peptides were purified by Prep-HPLC using a Reprosil-Pur C18 column (10 μm, 250 × 22 mm) eluted with water–acetonitrile gradients from 5% to 80% acetonitrile (0.1% formic acid, Sigma-Aldrich) in 30 min at a flowrate of 15.0 mL/min with UV detection at 220 nm. Mass spectrometry analysis was performed using a Bruker micrOTOF-Q instrument in positive mode to confirm the identity of the synthetic products (Appendix A). The epitope was derived from the hPG antigen with sequence the ATEGRVRVNSAYQDK. For coupling to FAM for flow cytometry and microscopy experiments, a lysine tetramer linker was added to the *N*-terminal of the sequence to compensate for the reduced solubility caused by dye conjugation, (i.e., hPG-FAM: FAM-KKKKATEGRVRVNSAYQDK).

### 2.2. Liposome Preparation

The phospholipids 1,2-distearoyl-*sn*-glycero-3-phosphocholine (DSPC) and 1,2-distearoyl-*sn*-glycero-3-phosphoglycerol (DSPG), were purchased from Avanti Polar Lipids, Birmingham,AL, USA. Cholesterol (CHOL) and RA were purchased from Sigma-Aldrich. Liposomes were prepared using the thin film dehydration–rehydration method, as described previously [25]. Briefly, phospholipids and CHOL (40 μmol, 4 mL) were dissolved in chloroform and methanol 1:1 and mixed in a 100 mL round-bottom flask at a molar ratio of 4:1:2DSPC:DSPG:CHOL. To prepare RA-encapsulating liposomes, 60 nmol of RA was added in this step. To prepare fluorescently labeled liposomes, 0.02 mol% of total lipid of 1,1′-dioctadecyl-3,3,3′,3′-tetramethylindodicarbocyanine, 4-chlorobenzenesulfonate salt (DiD, ThermoFisher) was added in this step. The solvents were evaporated under vacuum in a rotary evaporator for 1 h at 40 °C. The resulting lipid film was rehydrated with 1000 μg of hPG, hPG-K4 or hPG-K4-FAM dissolved in 4 mL of 10 mM 4-(2-hydroxyethyl)-1-piperazineethanesulfonic acid (HEPES, pH 7.2) buffer and homogenized by rotation in a water bath at 40 °C for 1 h. The multilamellar vesicle suspension was sized by high-pressure extrusion (LIPEX Extruder, Northern Lipids Inc., Burnaby, BC, Canada) by passing the dispersion four times through stacked 400 nm and 200 nm pore size membranes (Whatman^®^ NucleoporeTM, GE Healthcare, Amersham, UK). To separate non-encapsulated cargo from the liposomes, liposomes were pelleted by ultracentrifugation (Type 70.1 Ti rotor) for 50 min at 55,000 rpm at 4 °C. This was repeated three times. Liposomes were stored at 4 °C and their stability was measured periodically. Liposomes were determined to be unchanged for up to at least 1 year. Liposomes were used within 2 months for in vitro experiments and within 2 weeks for in vivo experiments.

### 2.3. Liposome Characterization

The Z-average diameter and polydispersity index (PDI) of the liposomes were measured by dynamic light scattering (DLS) using a NanoZS Zetasizer (Malvern Ltd., Malvern, UK). ζ-potential was measured by laser Doppler electrophoresis (Malvern Ltd.). For this, the liposomes were diluted 100-fold in HEPES buffer pH 7.2 to a total volume of 1 mL. To determine the concentration of loaded hPG and RA, the content of the liposomes was measured using RP-UPLC. For this, 10 uL of liposome suspension was dissolved in 190 uL of methanol, and the sample was vortexed. Sample injections were 5 μL in volume and the column used was a 1.7 μm BEH C18 column (2.1 × 50 mm, Waters ACQUITY UPLC, Waters, MA, USA). Column and sample temperatures were 40 °C and 20 °C, respectively. The mobile phases were Milli-Q water with 0.1% TFA (solvent A) and acetonitrile with 0.1% TFA (solvent B). For separation, the mobile phases were applied in a linear gradient from 5% to 95% solvent B over 6.5 min at a flow rate of 0.25 mL/min. hPG was detected by absorbance at 280 nm using an ACQUITY UPLC TUV detector (Waters ACQUITY UPLC, Waters, MA, USA). RA was detected at 351 nm.

### 2.4. Mice

Female Balb/cAnNCrl mice were purchased from Charles River Laboratories (Freiburg, Germany). Female mice were used as they are more susceptible to develop arthritis in the proteoglycan-induced arthritis model [26]. hPG T-cell receptor (TCR) transgenic Thy1.1^+^ mice [27] were bred in-house at Utrecht University under standard laboratory conditions. Mice were provided with food and water ad libitum.

### 2.5. Bone Marrow-Derived DC (BMDC) Culture

Penicillin, streptomycin and β-mercaptoethanol were purchased from Gibco (ThermoFisher, Landsmeer, Netherlands). Bone marrow was obtained from the femurs and tibias of Balb/cAnNCrl mice. Cells were cultured at 37 °C, 5% CO2 in a 6-well plate (Corning, Amsterdam, The Netherlands), at a density of 900,000 cells/mL, in IMDM (Gibco, Thermo Fisher Scientific), supplemented with 10% FCS (Bodinco, Alkmaar, The Netherlands), 100 units/mL of penicillin, 100 ug/mL of streptomycin and 0.5 μM β-mercaptoethanol in the presence of 20 ng/mL of granulocyte-macrophage colony-stimulating factor (GM-CSF, in house produced). Fresh medium and GM-CSF was added on day 2, and extra GM-CSF was supplemented to the culture on day 5. On day 7, cells were matured in the presence of 10 ng/mL of lipopolysaccharide (LPS, O111:B4; Sigma-Aldrich) and treated with free or encapsulated RA and hPG antigen, or controls. After 16 h, DCs were harvested for phenotypic characterization by flow cytometry or microscopy, for co-culture with T cells, or for in vivo transfer in mice.

### 2.6. T Cell Isolation and Co-Culture with BMDCs

Single-cell suspensions of splenocytes were obtained by mashing spleens of hPG TCR transgenic mice through a 70 μM filter, and erythrocytes were lysed with Ammonium–Chloride–Potassium (ACK) lysis buffer (0.15 M NH_4_Cl, 1 mM KHCO_3_, 0.1 mM Na_2_EDTA; pH 7.3). Subsequently, CD4^+^ T cells were negatively selected using Dynabeads™ (sheep anti-rat IgG, ThermoFisher) and anti-CD8 (YTS169), anti-CD11b (M1/70), anti-MHCII (M5/114) and anti-B220 (RA3-6B2) as described previously [12]. Enriched CD4^+^ T cells were labeled with carboxyfluorescein succinimidyl ester (CFSE, 0.5 nM) according to the manufacturer’s protocol (ThermoFisher). Selected T cells were co-cultured in a 2:1 ratio with DCs for 3 days at 37 °C and 5% CO_2_ and subsequently harvested for phenotypic characterization.

### 2.7. Adoptive Transfer of hPG TCR-Specific T Cells

CD4^+^CD25^−^ T cells were magnetically isolated from spleens of hPG TCR transgenic Thy1.1^+^ Balb/cAnNCrl mice as described above, with the addition of anti-CD25 (PC61, in-house produced) to the antibody mix to deplete activated T cells [28]. Enriched CD4^+^CD25^−^ T cells were labeled with 0.5 nM CFSE (ThermoFisher), resuspended in 200 uL of phosphate-buffered saline (PBS), and 300,000 cells were injected intravenously via the tail vein into wildtype Balb/cAnNCrl acceptor mice within 1 h. After 24 h, 1 × 10^6^ DCs pulsed with RA and hPG (tolDCs), free RA and hPG or liposomes encapsulating RA and hPG were injected into the acceptor mice. The concentrations of hPG and RA were 1 nmol and 0.2 nmol, respectively. After 72 h, acceptor mice were sacrificed, and spleens were harvested.

### 2.8. Flow Cytometry

For all flow cytometry experiments, the cell suspension was first blocked with Fc Block (2.4G2, in house produced). Extracellular staining was performed with a cocktail of antibodies in FACS Buffer (1X PBS supplemented with 2% FCS). For intracellular staining, the FoxP3 transcription factor staining set was used (eBioscience, San Diego, CA, USA). For all flow cytometric analyses, appropriate single-stain and fluorescence minus one controls were used. Flow cytometry was performed using the Beckman Coulter Cytoflex LX at the Flow Cytometry and Cell Sorting Facility at the Faculty of Veterinary Medicine at Utrecht University. Acquired data were analyzed using FlowJo Software v.10.7 (FlowJo LLC, Ashland, OR, USA). 

On day 8, BMDCs were stained with monoclonal antibodies CD11c-APC (N418, eBioscience, Thermo Fisher Scientific), MHCII-eFluor450 (M5/114.15.2, eBioscience, Thermo Fisher Scientific), CD40-PE (3/23, BD Biosciences, Franklin Lakes, NJ, USA), CD86-FITC (GL-1, BD Biosciences) and ViaKrome808 (Beckman Coulter, Indianapolis, IN, USA).

For the co-cultures, CD4+ T cells were harvested and transferred to a 96-well V-bottom plate for staining. T cells were stained with monoclonal antibodies CD4-BV785 (RM4-5, BioLegend, San Diego, CA, USA), CD25-PerCPCy5.5 (PC61.5, eBioscience, ThermoFisher, Landsmeer, Netherlands), CD49b-APC (DX5, BioLegend, USA), Lag-3-PE (C9B7W, eBioscience, Thermo Fisher Scientific), FoxP3-eFluor450 (FJK-16s, eBioscience, Thermo Fisher Scientific) and ViaKrome808 (Beckman Coulter, Indianapolis, IN, USA).

For adoptive T-cell transfer experiments, spleens of acceptor mice were harvested 72 h post-treatment, as described above. Spleens were mashed through a 70 μM filter and erythrocytes were lysed with ACK lysis buffer. Acquired splenocytes were stained with monoclonal antibodies CD4-BV785 (RM4-5, BioLegend, San Diego, CA, USA), Thy1.1 (CD90.1)-PerCP-Cy5.5 (HIS51, eBioscience, Thermo Fisher Scientific), CD44-APC (IM7, eBioscience, Thermo Fisher Scientific), CD62L-PE (MEL-14, BD Biosciences), CTLA-4 (CD152)-BV605 (UC10-4B9, eBioscience, Thermo Fisher Scientific), FoxP3-eFluor450 (FJK-16s, eBioscience, Thermo Fisher Scientific), RORγT-PE (AFKJS-9, eBioscience, Thermo Fisher Scientific), GATA-3-PE-Cy7 (TWAJ, eBioscience, Thermo Fisher Scientific), T-Bet-APC (4B10, eBioscience, Thermo Fisher Scientific), CD11c-APC (N418, eBiosciences, Thermo Fisher Scientific), CD11b-PerCP-Cy5.5 (M1/70, eBioscience, Thermo Fisher Scientific), MHCII(I-A/I-E)-PE-Cy5 (M5/114.15.2, eBioscience, Thermo Fisher Scientific), CD40-PE (3/23, BD Biosciences), CD8α-V500 (53–6.7, BD Biosciences) and ViaKrome808 Beckman Coulter, Indianapolis, IN, USA) in different flow cytometry panels to avoid spectral overlap.

### 2.9. Live Cell Imaging

On day 7, BMDCs were harvested and 75,000 cells per well were added to 35 mm glass-bottom cell culture dishes (CELLview™, Greiner Bio-One, Kremsmünster, Austria). Amounts of 1 μg/mL of hPG-FAM, free or in liposomes, or controls were added to the cells, together with 10 ng/mL of LPS. Cells were cultured overnight at 37 °C and 5% CO_2_. Before imaging, cells were carefully washed to remove unbound liposomes and/or hPG-FAM. An amount of 5 μg/mL of membrane permeable DNA stain Hoechst 33342 (Thermo Fisher) was added to each well shortly before imaging. Microscopy and analysis were performed at the Center for Cell Imaging at the Faculty of Veterinary Medicine at Utrecht University. Images were acquired using a NIKON A1R confocal microscope with a 40x Plan Apo objective (NA 1.3). Standard lasers and filter settings were used to detect Hoechst, FAM and DiD. Representative images were processed in NIS elements 5.02 (Nikon Microsystems, Europe).

## 3. Results

### 3.1. RA Induces a tolDC Phenotype in BMDCs In Vitro

tolDCs are generally described as having an immature phenotype and express the pan-DC marker CD11c, the antigen-presenting molecule MHCII, and have low expression of the co-stimulatory molecules CD40 and CD86. To assess whether the vitamin-A-derived RA could induce this tolDC phenotype in vitro, we tested different concentrations of RA on BMDCs. Cells were simultaneously incubated with LPS. The expression of each marker is presented as the geometric mean fluorescence intensity (MFI) and was determined by flow cytometry analysis. The addition of RA results in a significant decrease in the expression of MHC II (Figure 1A) and co-stimulatory molecules CD40 (Figure 1B) and CD86 (Figure 1C) on the cell surface of BMDCs. Increasing the dose of RA that was administered to the cells 10-fold did not result in significant reductions in the expression of the measured surface proteins compared to the lower concentrations of RA (Figure 1). The addition of RA to a BMDC culture therefore seems to be a promising way of inducing tolDCs.

### 3.2. RA Can Be Efficiently Encapsulated into Liposomes and Retains tolDC Inducing Effects

To assess whether RA remains able to induce tolDCs when encapsulated in liposomes, DSPC:DSPG:CHOL liposomes were loaded with hPG with or without RA. The liposomes were characterized using DLS and laser Doppler electrophoresis (Table 1). To assess liposome uptake by DCs in vitro, liposomes were fluorescently labeled using DiD and added to BMDCs. After 24 h of incubation, around 30% of all BMDCs were able to take up hPG-loaded liposomes or hPG/RA-loaded liposomes (Figure 2A). To address whether the liposomes were able to induce phenotypic tolDCs, BMDCs were stimulated in the presence of LPS with free hPG, free hPG and RA, hPG-loaded liposomes and hPG/RA-loaded liposomes. DCs treated with RA, free or in liposomes, show a reduced expression of CD86 (Figure 2B) and CD40 (Figure 2C) on their cell surface compared to the control hPG, indicating the inhibition of maturation. The expression of MHCII (Figure 2D) and PD-L1, involved in T cell suppression [29] (Figure 2E), seemed to remain similar for all DCs regardless of stimulation. The expression of chemokine receptor CCR7, essential for homing to secondary lymph nodes [30], was upregulated in tolDCs generated with free RA or RA liposomes (Figure 2F).

### 3.3. Liposomal Co-Delivery of hPG and RA Leads to Aantigen-Associated tolDC Induction In Vitro

To assess whether the liposomes affected antigen delivery, the hPG antigen was modified to include the fluorescent label FAM. Liposomes were prepared using hPG-FAM and this modification did not affect liposomal properties (Appendix A). BMDCs were incubated with LPS in the presence of hPG-FAM, hPG-FAM and RA, hPG-FAM liposomes or hPG-FAM/RA liposomes for 24 h. The free hPG-FAM readily associated with cells (Figure 3C), and microscopy showed that most of the antigen was located at the surface of the BMDCs (Figure 4). In contrast, while flow cytometry showed that fewer cells were positive for the FAM label when the antigen was encapsulated in liposomes (Figure 3C), microscopy showed that the FAM label was mostly present inside the cells (Figure 4 and Appendix A). Interestingly, the presence of RA reduces the uptake of the antigen (Figure 3C). Next, we aimed to determine whether antigen-associated cells were also phenotypically tolDCs. For this, the expression of CD86 and CD40 was measured in the hPG-FAM^+^ DCs. hPG-FAM liposomes and hPG-FAM/RA liposomes induced more tolDCs (as defined by CD40^−^ or CD86^−^CD11c^+^ DCs) than free hPG-FAM or free hPG-FAM and RA within antigen-associated cells (Figure 3A,B). Interestingly, only the free hPG-FAM and RA increased non-antigen-loaded tolDCs (Figure 3D,E).

### 3.4. tolDCs Generated with hPG/RA Liposomes Skew T Cells towards a Regulatory Phenotype In Vitro

hPG/RA liposomes can inhibit CD40 and CD86 expression in DCs to the same extent as free hPG and RA, but the question about the functionality of these DCs remains. To assess the ability of these liposome-induced tolDCs to reduce effector T cells (Teff) and stimulate regulatory T-cell (Treg) proliferation, purified hPG-TCR transgenic CD4^+^ T cells were co-cultured with DCs pulsed with different conditions. tolDCs generated using free hPG and RA and DCs that were induced through hPG/RA liposome stimulation induced significantly less CD4^+^ T cell proliferation compared to the pro-inflammatory mDCs control (Figure 5A). All groups induced Tregs (Figure 5B), as shown by the increase in the expression of CD25+ FoxP3+ T cells. Additionally, reduced populations of Tbet+ Teffs (Figure 5C) were observed in all treatments compared to the control group, suggesting a decrease in the inflammatory Th1 cell population.

### 3.5. hPG and RA Delivered by tolDCs, Liposomes or Free Affect Splenic CD11c^+^ Cell Populations In Vivo

To observe whether hPG and RA-induced tolDCs, hPG/RA liposomes or free hPG/RA had effects on splenic DC population phenotype, mice were injected intravenously with the different formulations. Three days after injection, we characterized CD11c^+^ cells in the spleen using flow cytometry. Within splenic CD11c^+^ cells, we found no differences in the % of CD11b^hi^ and MHC-II^lo^ cells (Figure 6B,C). However, %CD8α^+^CD11c^+^ cells were significantly increased in mice receiving tolDCs compared to mice receiving hPG/RA liposomes (Figure 6D). Furthermore, mice receiving the liposomes had significantly fewer MHC-II^hi^CD40^hi^-activated cells compared to mice that received free hPG/RA (Figure 6E).

### 3.6. hPG and RA Delivered by tolDCs, Liposomes or Free Affect Splenic CD4^+^ T Cell Populations In Vivo

To assess the effect on antigen specific T-cell responses by hPG and RA administration, we performed an in vivo adoptive transfer study. Mice received CFSE-labeled Thy1.1^+^CD4^+^CD25^−^ T cells isolated from hPG-TCR transgenic mice, followed by an intravenous injection of tolDCs, hPG/RA liposomes or free hPG/RA. Both the Thy1.1^+^ hPG-specific and the bystander Thy1.1^−^ CD4^+^ T cell populations were evaluated by flow cytometry. Strikingly, tolDCs showed the highest activation of antigen-specific CD4^+^ T cells, as measured by % Thy1.1^+^ cells, % CFSE^−^ cells and % CD25^+^ cells (Figure 7A–C). This proliferation was due to the presence of the antigen, since mice that received only PBS showed no or hardly any CFSE- Thy1.1+ CD4+ T cells (data not shown). Within memory T-cell subsets, there were no differences between the groups within the antigen-specific T-cell populations (Figure 7D–F). However, in the Thy1.1^2−^ population, tolDCs reduced the % of naïve CD4^+^ T cells compared to the other groups (Figure 7D). Within the antigen-specific immune cell subsets, there were no differences in the induction of CD25^+^FOXP3^+^ (Treg), T-bet^+^ (Th1), GATA-3^+^ (Th2), and RORγT^+^ (Th17) CD4^+^ T cells (Figure 8A–D). However, the %CTLA-4^+^ cells were significantly lower in the tolDC group compared to the other groups (Figure 8E). CTLA-4 is a negative regulator of T cell activation [31]. Comparing the Thy1.1^+^ vs. Thy1.1^−^ effects within each group, we observed increased CD25^+^FOXP3^+^ and CTLA-4^+^ cells regardless of delivery method (Figure 8A,E), and tolDCs reduced T-bet^+^ cells (Figure 8B). Interestingly, bystander RORγT cells were significantly higher in mice receiving tolDCs compared to mice receiving liposomes (Figure 8D).

## 4. Discussion and Conclusions

The restoration of antigen-specific tolerance is essential for the development of a curative therapy for autoimmune diseases. tolDCs have shown promising results in the induction of antigen-specific tolerance in animal models and positive results in clinical trials [22]. Here, we focused on the naturally occurring tolerance-inducing compound RA and evaluated several methods for delivering RA to dendritic cells in vitro and in vivo. In vitro, we show that RA can induce a semi-mature phenotype that is characteristic of tolDCs (Figure 1 and Figure 2), which is in line with other studies [14,15,16]. These tolDCs were functional and could inhibit antigen-specific T-cell proliferation while increasing the relative population of Tregs and reducing the Th1 population (Figure 5). Uptake studies with fluorescently labeled hPG-FAM in BMDCs revealed that the incubation of these cells with free hPG-FAM and RA leads to a heterogeneous population (Figure 3), suggesting that not all cells that take up antigens also become tolerogenic and vice versa. When this heterogeneous population of cultured RA-tolDCs was injected intravenously in mice, we observed not only antigen-specific effects, but also found changes in non-antigen-specific T-cell subsets (Figure 7). We also studied several subsets of splenic DCs that might give more insight into the general immune environment of the spleen after the different treatments. We found no differences in the immunosuppressive CD11b^+^CD11c^+^ population [32]; however, the injection of tolDCs increased the proportion of CD8α^+^CD11c^+^ cells in the spleens of mice (Figure 6D). CD8α^+^ DCs have been described to take up apoptotic cells in lymphoid tissues and are highly efficient at cross-presentation in MHC-I [33], which is important for the induction of CD8 T cells. CD8α^+^ DCs are also considered to be vital for maintaining immune tolerance [34,35]; however, there is a report of these cells accelerating the progression of collagen-induced arthritis in mice [36], which is a murine model for autoimmune disease. Further studies on the involvement of different DC subsets in the regulation of autoimmune diseases are needed to clarify the role of these cells in tolerance induction. To mitigate the observed effect on non-antigen-specific T cells, we theorized that co-delivery of the antigen and RA by a nanoparticle, such as a liposome, would be a better strategy for inducing antigen-specific tolerance and limiting off-target effects.

The liposomes we selected have been previously shown to induce Tregs in vivo [25,37], but their effect on DCs had not been studied. BMDCs incubated with LPS and hPG behaved similarly to cells incubated with LPS and hPG-containing liposomes (Figure 2). These hPG-containing liposomes only had a small effect on CD4^+^ T-cell proliferation in a co-culture assay (Figure 5A). However, we did observe an in vitro effect of liposome-pulsed BMDCs on the induction of antigen-specific Tregs and Th1 cells (Figure 5B,C). Furthermore, after in vitro incubation with BMDCs, we saw striking differences between hPG-FAM that was given freely to these cells and hPG-FAM encapsulated in liposomes (Figure 3 and Figure 4). The liposomal delivery of the antigen reduced the % of cells which had taken up antigens from 69 ± 5% to 8 ± 1% (*p* < 0.0001), which was possibly due to the cationic charge of the antigen (isoelectric point 10.88) compared to the anionic charge of the liposomes [38]. Even without the addition of RA, the hPG-containing liposomes themselves can induce antigen-specific tolDCs and Tregs (Figure 5). This was also observed in previous studies whereby antigen-loaded DSPC:DSPG:CHOL liposomes induced antigen-specific Tregs and mitigated the progression of atherosclerosis in mice [25]. While this is promising, we hypothesized that the addition of RA in the hPG liposomes would enhance tolerance induction even further.

The co-encapsulation of RA and hPG in liposomes did not alter their physicochemical properties as compared to hPG alone (Table 1). This is likely because the hPG is localized in the aqueous core of the liposomes, while the hydrophobic RA is incorporated into the lipid bilayer of the liposomes. RA induced tolDCs equally efficiently when given freely to BMDCs or when encapsulated in liposomes (Figure 2). This suggested that while liposomes do not affect the ability of RA to induce tolDCs, they had no advantage over free RA. This was also reflected in in vitro co-culture assays with antigen-specific CD4^+^ T cells; free RA had the same effects as RA encapsulated in liposomes (Figure 5). Capurso et al. similarly observed an equivalent effect between free RA or RA encapsulated in poly(lactic-co-glycolic acid) nanoparticles [39]. These results are expected since the in vitro system cannot reproduce the complex parameters that direct the (co-)delivery of compounds to APCs in vivo, such as administration route, clearance rate, biodistribution and stability. Within the antigen-associated subset of cells, the liposomal delivery of RA led to a significantly higher proportion of tolDCs in vitro as compared to free RA (Figure 3A,B), while the opposite was observed in non-antigen-specific tolDCs (Figure 3D,E). Comparing hPG/RA liposomes with hPG liposomes, the hPG/RA liposomes caused a larger decline in proliferated CD4^+^ T cells (Figure 5). In the in vivo experiment, the liposomes had the lowest %MHC-II^hi^CD40^hi^ cells in the CD11c^+^ population in the spleen, suggesting that the liposomes are (indirectly) interacting with splenic CD11c^+^ cells to inhibit their activation. In addition, the injection of hPG/RA liposomes mitigated bystander effects in non-antigen-specific T cells (Figure 7 and Figure 8) but had no effect compared to the other groups on antigen-specific T cells. Similarly, Phillips et al. found that, after the subcutaneous injection of microparticles consisting of human insulin peptide B_9-23_, RA and TGF-β, there was no change in Tregs in mice compared to controls. However, they did find a significant increase in regulatory B cells in the mesenteric lymph nodes 3 days after microparticle injection, and the mice in this group had a significant reduction in diabetes progression [40]. These data combined show that nanoparticle delivery of RA can be a more specific method to induce antigen-specific tolerance compared to tolDCs.

While there were some differences in T cell subsets between the groups in vivo, it should be stressed that the proportion of antigen-specific Tregs in all mice were significantly enhanced compared to the background and effector T cells (Figure 8). While Treg induction was the main goal of the current study, it would be interesting to further study the mechanisms whereby tolDCs and liposomes induce tolerance in more detail. While our hPG-FAM experiments do give some insights into this, the effect will likely be different in an in vivo system. This could be achieved by injecting fluorescently labeled (antigens in) tolDCs and liposomes and tracking their biodistribution over time. Tracking tolDCs or liposomes would also give more information about the in vivo phenotypical stability of the tolDCs and the phenotype of antigen- or liposome-associated cells in vivo. Finally, other dosage schemes or administration routes could improve the effects of RA liposomes on Treg induction.

In conclusion, we show that RA is a potent immunomodulator for the induction of antigen-specific tolerance and that DSPC:DSPG:CHOL liposomes are a suitable carrier system for the co-delivery of an antigen with RA in vitro. Additionally, we show the strong induction of antigen-specific Tregs, with no off-target effects when using these liposomes. Although the in vitro data seems very promising, generating the same effects in vivo remains challenging. In this work, we looked at the heterogeneous populations of DCs that arise in a tolDC culture in vitro and the bystander effect of immunosuppressive therapy in vivo. This stresses the importance of not only measuring antigen-specific effects, but also considering off-target effects. The optimization of in vivo administration and thorough examination of off-target effects of RA-tolDC or RA-liposome treatment could provide new insights in the use of RA for antigen-specific immunotherapy for autoimmune and chronic inflammatory diseases.

## Figures and Tables

**Figure 1 pharmaceutics-13-01949-f001:**
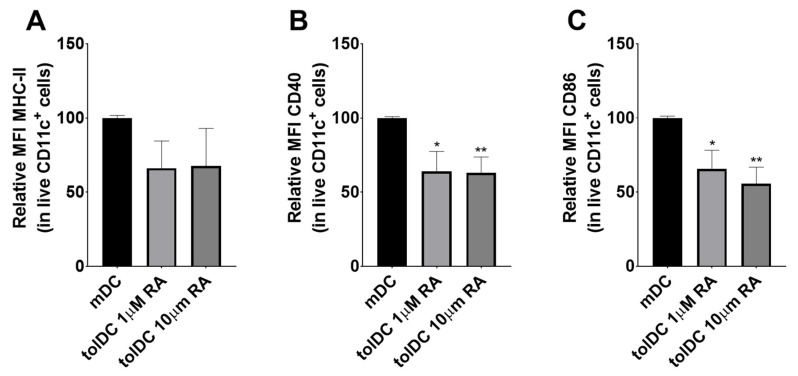
Addition of RA results in a tolDC phenotype in vitro. BMDCs were cultured from the bone marrow of Balb/c mice. BMDCs were stimulated with LPS (mDC) or LPS and different concentrations of RA (tolDC; 1 or 10 uM RA). The concentration of LPS was constant for all groups. After 24 h of incubation, cells were washed thoroughly to remove stimuli and analyzed via flow cytometry. Relative MFI normalized to mDC of (**A**) MHCII expression (**B**) CD40 expression, and (**C**) CD86 expression on live CD11c^+^ BMDCs. Combined data of three independent experiments. *n* = 3. Means + SD, * *p* < 0.05, ** *p* < 0.01 compared to mDC, as determined by one-way ANOVA and Tukey’s multiple comparisons test.

**Figure 2 pharmaceutics-13-01949-f002:**
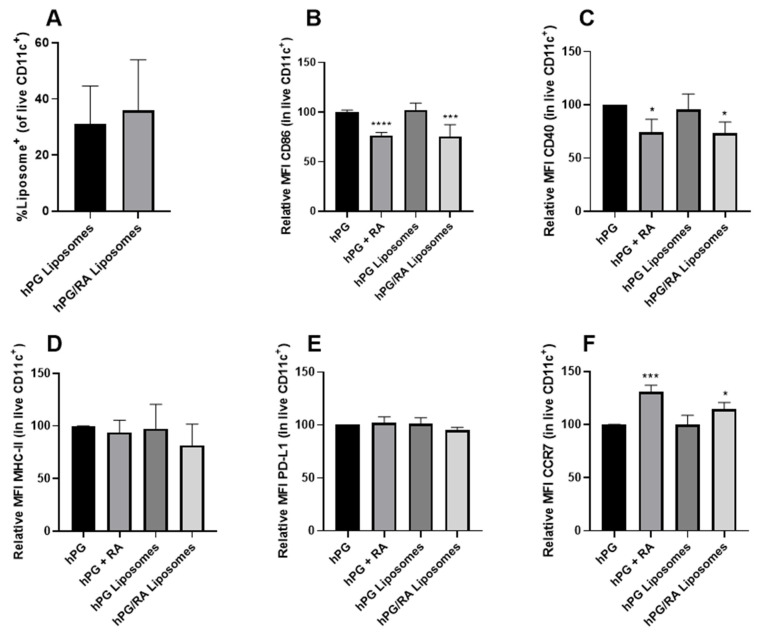
RA, free or encapsulated in liposomes, induces a tolDC phenotype in BMDCs. BMDCs were stimulated with LPS and cultured in the presence of hPG, hPG + RA, hPG liposomes or hPG/RA liposomes. The hPG and RA concentrations were constant in all groups. After 24 h of incubation, cells were washed thoroughly to remove unbound liposomes and analyzed via flow cytometry. (**A**) Percentage of live BMDCs positive for the fluorescent label in the liposomes. (**B**–**F**) Relative MFIs (compared to hPG control) of several DC markers. Combined data of four independent experiments. Means + SD, **** *p* < 0.0001, *** *p* < 0.001, * *p* < 0.05 compared to free hPG determined by mixed-effects analysis and Dunnett’s multiple comparisons test.

**Figure 3 pharmaceutics-13-01949-f003:**
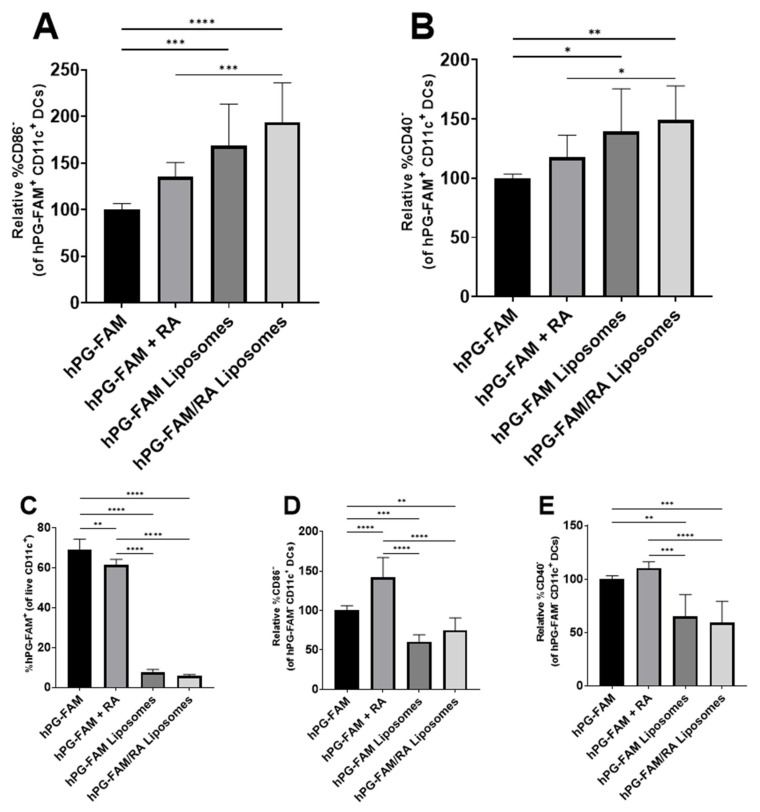
Encapsulation of hPG in liposomes skews towards tolDCs within antigen-associated BMDCs. BMDCs were stimulated with LPS and cultured in the presence of hPG-FAM, hPG-FAM + RA, hPG-FAM liposome or hPG-FAM/RA liposomes. After 24 h of incubation, cells were washed thoroughly and analyzed by flow cytometry. (**A**) Relative % CD86^−^ (normalized to hPG-FAM control) and (**B**) CD40^−^ tolDCs within hPG-FAM^+^ cells. (**C**) Percentage of live BMDCs positive for FAM. (**D**) Relative %CD86^−^ (normalized to hPG-FAM control) and (**E**) CD40^−^ tolDCs within hPG-FAM^−^ cells. Means + SD, **** *p* < 0.0001, *** *p* < 0.001, ** *p* < 0.01, * *p* < 0.05 compared to hPG-FAM determined by one-way ANOVA and Tukey’s multiple comparisons test. The data shown are combined normalized data from three independent experiments.

**Figure 4 pharmaceutics-13-01949-f004:**
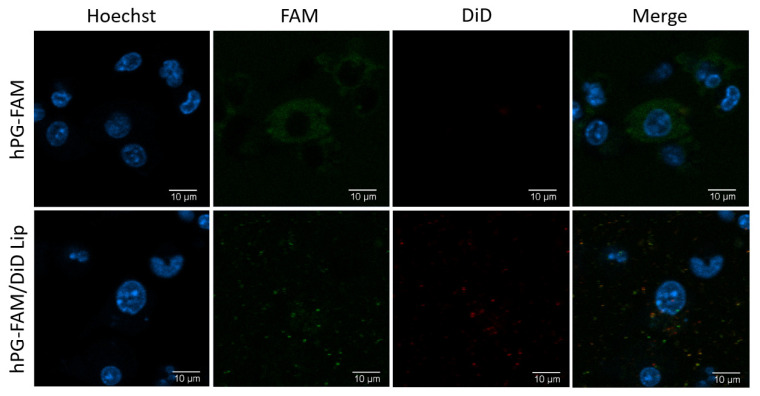
Peptide encapsulation in liposomes enhances uptake by BMDCs in vitro. BMDCs were stimulated with LPS and cultured in the presence of hPG-FAM added freely or encapsulated in DiD-labeled liposomes. After 24 h of incubation, cells were washed to remove unbound antigen and liposomes. The blue signal shows the Hoechst staining, the green signal indicates the presence of hPG-FAM and in red is the liposomal dye. The scale bar shows 10 μm. *N* = 1.

**Figure 5 pharmaceutics-13-01949-f005:**
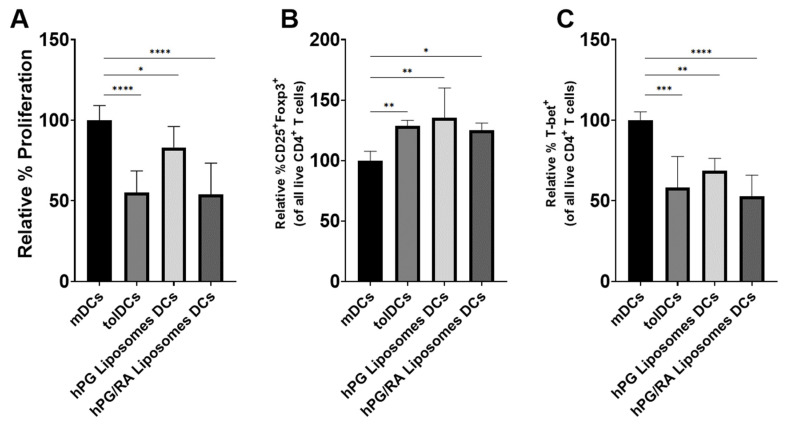
RA, free or encapsulated in liposomes, skew T cells towards a regulatory phenotype. BMDCs were stimulated with LPS and cultured in the presence of hPG (mDCs), hPG + RA (tolDCs) or hPG-containing liposomes with or without RA (hPG liposome DCs and hPG/RA liposome DCs, respectively). The hPG and RA concentrations were constant in all groups. After 24 h of incubation, the BMDCs were washed and hPG-TCR-specific CD4^+^ T cells were added. T-cell activation was analyzed by flow cytometry after 72 h of co-culture. (**A**) Relative % proliferated cells compared to mDC control. (**B**) Relative % CD25^+^Foxp3^+^ and (**C**) % T-bet^+^ in all CD4^+^ T cells, normalized to mDCs. Means + SD, **** *p* < 0.0001, *** *p* < 0.001, ** *p* < 0.01, * *p* < 0.05 compared to mDC determined by one-way ANOVA and Tukey’s multiple comparisons test. The data shown are combined normalized data from three independent experiments.

**Figure 6 pharmaceutics-13-01949-f006:**
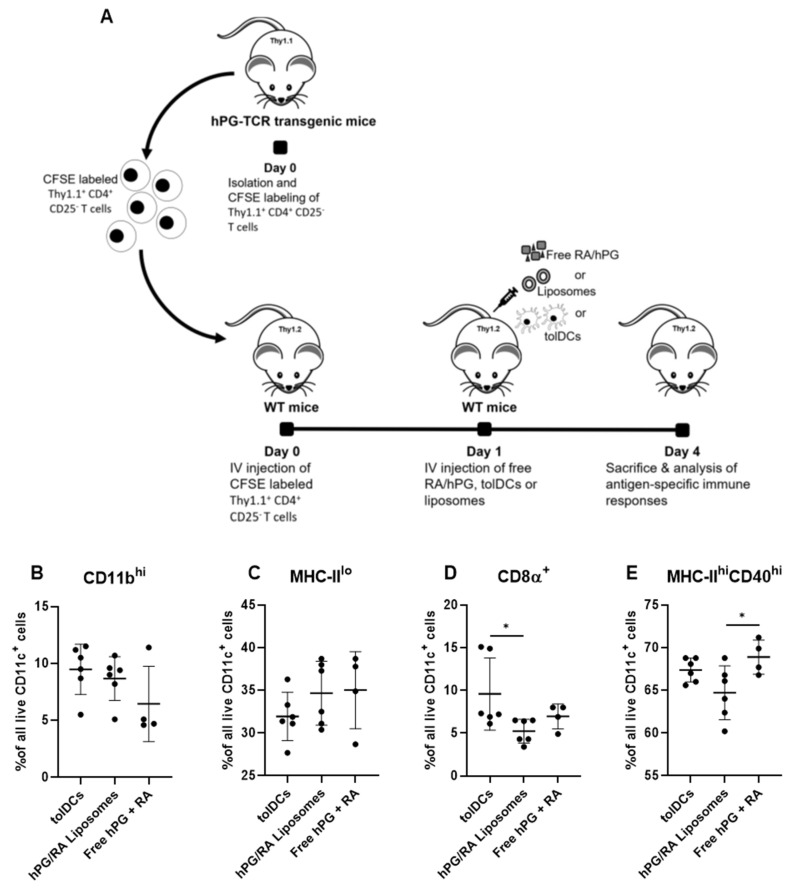
Effect of RA and hPG delivery on splenic DCs in vivo. (**A**) After adoptive transfer of Thy1.1^+^CD4^+^CD25^−^ T cells, mice received intravenous injections of either tolDCs pulsed with hPG + RA (tolDCs), liposomes encapsulating hPG + RA (hPG/RA liposomes), or free hPG + RA (free hPG + RA). Three days after injection, splenic DC populations (in live CD11c^+^ cells) were assessed by flow cytometry. The % of (**B**) CD11b^hi^ DCs, (**C**) MHC-II^lo^ DCs, (**D**) CD8α^+^ DCs and (**E**) MHC-II^hi^CD40^hi^ DCs were determined. *n* = 6 for tolDCs and hPG/RA Liposomes, *n* = 4 for free hPG + RA control. Means ± SD, * *p* < 0.05 determined by one-way ANOVA and Tukey’s multiple comparisons test.

**Figure 7 pharmaceutics-13-01949-f007:**
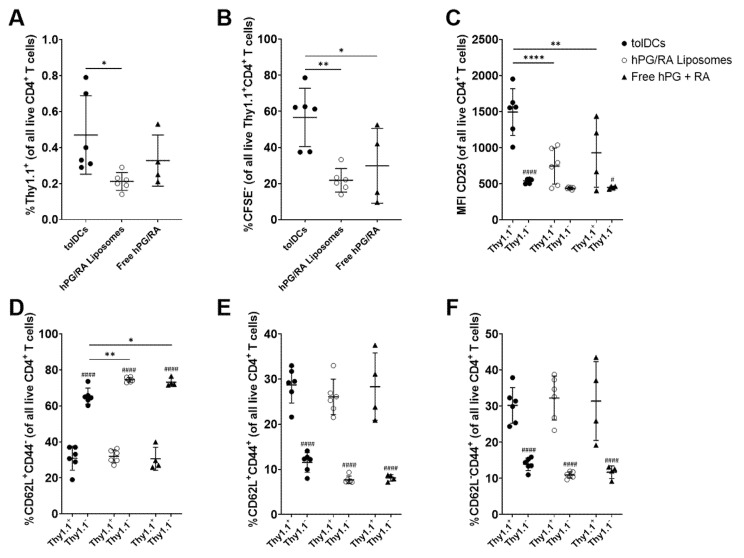
Effect of RA and hPG delivery on the activation of splenic CD4+ T cells in vivo. WT Balb/c mice received CFSE-labeled Thy1.1^+^CD4^+^CD25^−^ T cells isolated from the spleens of hPG-TCR transgenic mice via intravenous injection. 24 h after injection, mice received intravenous injections of either tolDCs pulsed with hPG and RA (tolDCs), liposomes encapsulating hPG and RA (hPG/RA Liposomes), or free hPG + RA. Three days after injection, splenic CD4^+^ T cells were assessed by flow cytometry. (**A**) % of antigen-specific Thy1.1^+^ CD4^+^ T cells, and (**B**) % of proliferated CFSE^−^ cells within this population. (**C**) CD25^+^, (**D**) naïve CD62L^+^CD44^−^, (**E**) central memory CD62L^+^CD44^+^, (**F**) and effector CD62L^−^CD44^+^ cells within the Thy1.1^+^ and Thy1.1^−^ CD4^+^ T cell populations. *n* = 6 for tolDCs and hPG/RA Liposomes, *n* = 4 for free hPG + RA control. Means ± SD, **** *p* < 0.0001, ** *p* < 0.01, * *p* < 0.05. #### *p* < 0.0001, # *p* < 0.05 comparing Thy1.1^+^ to Thy1.1^−^. Statistics were performed by one-way or two-way ANOVA and Bonferroni’s multiple comparisons test.

**Figure 8 pharmaceutics-13-01949-f008:**
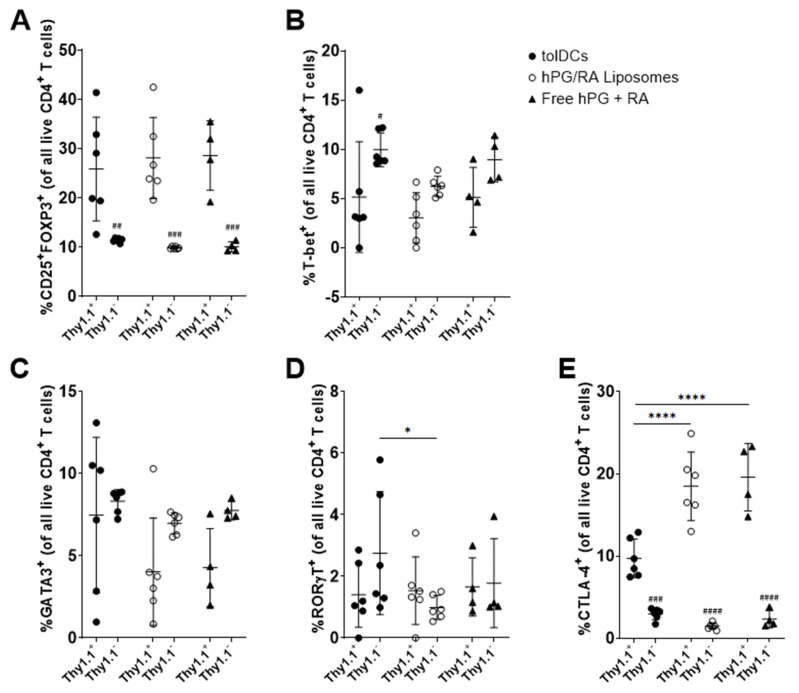
Effect of RA and hPG delivery on the splenic CD4^+^ T cell subsets in vivo. WT Balb/c mice received CFSE-labeled Thy1.1^+^CD4^+^CD25^−^ T cells isolated from the spleens of hPG-TCR transgenic mice via intravenous injection. 24 h after injection, mice received intravenous injections of either tolDCs pulsed with hPG and RA (tolDCs), liposomes encapsulating hPG and RA (hPG/RA Liposomes), or free hPG + RA. Three days after injection, splenic CD4^+^ T cells were assessed by flow cytometry. (**A**) %CD25^+^FOXP3^+^ (**B**) %T-bet^+^ (**C**) %GATA3^+^, (**D**) %RORγT^+^ (**E**) and % CTLA-4^+^ cells within the Thy1.1^+^ and Thy1.1^−^ CD4^+^ T cell populations. *n* = 6 for tolDCs and hPG/RA Liposomes, *n* = 4 for free hPG + RA control. Means ± SD, **** *p* < 0.0001, * *p* < 0.05. #### *p* < 0.0001, ### *p* < 0.001, ## *p* < 0.01, # *p* < 0.05 comparing Thy1.1^+^ to Thy1.1^−^. Statistics performed by two-way ANOVA and Bonferroni’s multiple comparisons test.

**Table 1 pharmaceutics-13-01949-t001:** Properties of liposomes, means ± SD.

Formulation	Z-Average Diameter (nm)	ζ-Potential (mV)	PDI	Encapsulation hPG (%)	Encapsulation RA (%)
hPG	186.8 ± 11.2	−47.7 ± 2.1	0.10 ± 0.05	57.3 ± 3.3	-
hPG/RA	183.7 ± 4.9	−45.9 ± 0.9	0.07 ± 0.01	43.9 ± 4.5	79.5 ± 29.0

## Data Availability

The data presented in this study are available on request from the corresponding author.

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
