# Peer review of "Retinoic Acid-Containing Liposomes for the Induction of Antigen-Specific Regulatory T Cells as a Treatment for Autoimmune Diseases"

_pharmaceutics, 2021, doi:10.3390/pharmaceutics13111949_

Round 1

Reviewer 1 Report

The article represents an in vitro study of tolerogenic murine dendritic cells produced by means of liposomes produced with proteoglycan-derived peptide and differentiated by retinoic acid, being able to induce T-regulatory cell response.   An effort was made to make the population more homogenous by means of liposomes carrying a complex lipid component (DSPG), thus sufficiently increasing their contact with antigen, and tolerogenic properties of these cells directed to T-reg cells.  The in vitro and murine experiments are well designed and described. The liposome preparations are well characterized, with respect to size and potential and encapsulation efficiency. The T-reg induction mediated by dendritic tolerogenic cells as well as potentiating effect of  all-trans-retinoic acid are well confirmed.

There are no principal arguments against publication of this paper. The quite artificial in vitro model results confirm the previously obtained data, thus providing expected positive results.

The article could be published without sufficient changes.

Author Response

Dear reviewer,

Thank you for the positive evaluation of our manuscript.

Reviewer 2 Report

REVIEWER’S COMMENTS

The manuscript Retinoic Acid-containing Liposomes for the Induction of Antigen-specific Regulatory T Cells as a Treatment for Autoimmune Diseasesby Braake et al, highlights the importance of co-delivery of an antigen and immunomodulator to minimize off-target effects and systemic side-effects and provides new insights in the use of RA for antigen-specific immunotherapy for autoimmune and chronic inflammatory diseases.

  1. What experiments would the authors perform for optimization of in vivo RA administration? Please elaborate in the discussion section.
  2. Why were only female Balb/cAnNCrl mice used in this study. Is there an immune component involved that differs with gender?
  3. Please include the exact p-values and mean ± SEM values in the results section.
  4. Please include the catalog numbers and manufacturer of the assay kits, reagents, and antibodies used in this study.
  5. Please include the dilution/ concentration of primary and secondary antibodies used in this study.
  6. Please be consistent with the style of references.

Author Response

Dear reviewer,

Thank you for your feedback. Please find attached our answers to your comments. 

Kind regards,

Reviewer 3 Report

The manuscript discussed the effect of RA and hPG loaded liposomes on the antigen specific regulatory T cells and CD4+ cells in vitro and in vivo. They found liposomes can be promising in codeliver RA and hPG for potential use of autoimmune disease. However, we need to note:

  1. The liposomes seem not to improve the selectivity of the therapy in vivo, which is the claimed benefits of the whole manuscript compared to free drugs.
  2. Please also include how the antigens on the tolDC will help increase the selectivity in introduction.
  3. Please discuss how your method for ex vivo culture can overcome which problems that are mentioned in line 76-79. It looks like you still need to extract the cells and go through all the process here.
  4. In all figures, please indicate n equals what in each group in the legend and description.
  5. In figure 1, to what extent can we define it tolDC, rather than mDC. Thus, it is better to have a positive control for tolDC.
  6. In figure 1, please draw error bar in control group (LPS), now that you have it in fig 2.
  7. In figure 2, double check the data for fig 2B, fig 2C and fig 2F. It seems not significantly different among the groups. What parameters did you use for statistical analysis? Meanwhile, a question still needs to be addressed: the maturation level to what extent will be the characteristics of tolDC. Thus, a positive control for tolDC is very important, given the error bars are overlapped, especially in fig 2C. Next, a control for mDC without any treatments is also needed here to confirm the generation of mDC by hPG.
  8. In figure 3, the authors really need to check the statistical analysis in A and B, C between free hPG-FAM and freen hPG-FAM+RA.
  9. Why CD86 and CD40 are detected separately? Can tolDC be either CD86- or CD40-?
  10. In figure 4, DID +hPG-FAM was also needed to prove hPG-FAM was inserted in the membrane.
  11. In figure 5, check the statistical analysis of B between the first and 3rd group.
  12. In figure 5, it is also better to test if the loading of hPG can also decrease the antigen-unspecific T cell population to prove the improved selectivity by the formulations.
  13. Please elaborate what Thy1.1 and CD25 does. Though it can be comprehended, but it will take a lot of time to get what the authors try to prove.
  14. In figure 7 and 8, all the groups lack the key control which should not receive treatment, making the comparison much less reliable.

Author Response

Dear reviewer,

Thank you for your remarks on our manuscript. Please find attached our answers to your feedback.

Kind regards,

Round 2

Reviewer 3 Report

I think the manuscript has been greatly improved. But we still need to note:

  1. The statistical analysis is really a problem in figure 2 B, C. I do not understand how the authors can get significant difference by have the error bars overlap so much to each other. These data need to be carefully reviewed. A statistician is advised to help the authors.
  2. In my question 10, the group of DID+liposomes, serving as positive control is needed to help to prove the antigen is really embedded in the membrane,  not simply mixed them. 
